The impact of parental investment on lifetime reproductive success in Iceland

Lynch Robert F. robertflynch@gmail.com
Lynch Emily C.
Department of Anthropology, University of Missouri , Columbia , MO , United States
Calafell Francesc
Electronic publication date: 2017 Jan 19
Publication date: 2017
Volume: 5
Electronic Location ID: e2904
Received 2016 Sep 6; Accepted 2016 Dec 12
Copyright: ©2017 Lynch and Lynch
Copyright year: 2017
Copyright holder: Lynch and Lynch
License: This is an open access article distributed under the terms of the Creative Commons Attribution License, which permits unrestricted use, distribution, reproduction and adaptation in any medium and for any purpose provided that it is properly attributed. For attribution, the original author(s), title, publication source (PeerJ) and either DOI or URL of the article must be cited.
License URL: https://creativecommons.org/licenses/by/4.0/

Keywords: Parental investment, Reproductive success, Heritability, Human evolution

Funding: Fulbright Scholarship and Center of Human Evolutionary Studies Small Grant, Rutgers University This study was funded by the Fulbright Scholarship and Center of Human Evolutionary Studies Small Grant, Rutgers University. The funders had no role in study design, data collection and analysis, decision to publish, or preparation of the manuscript.

==============================
Background

Demonstrating the impact that parents have on the fitness of their children is a crucial step towards understanding how parental investment has affected human evolution. Parents not only transfer genes to their children, they also influence their environments. By analyzing reproductive patterns within and between different categories of close relatives, this study provides insight into the genetic and environmental effects that parents have on the fitness of their offspring.

Methods

We use data spanning over two centuries from an exceptionally accurate Icelandic genealogy, Íslendingabók, to analyze the relationship between the fertility rates of close relatives. Also, using genetic data, we determine narrow sense heritability estimates (h2) to further explore the genetic impact on lifetime reproductive success. Finally, we construct four simulations to model the expected contribution of genes and resources on reproductive success.

Results

The relationship between the reproduction of all full sibling pairs was significant and positive across all birth decades (r = 0.19) while the reproductive relationship between parents and offspring was often negative across many decades and undetectable overall (r = 0.00) (Fig. 1 and Table 1). Meanwhile, genetic data among 8,456 pairs of full siblings revealed a narrow sense heritability estimate (h2) of 0.00 for lifetime reproductive success. A resources model (following the rule that resources are transmitted from parents to children, distributed equally among siblings, and are the only factor affecting reproductive success) revealed a similar trend: a negative relationship between parent and offspring reproduction (r =  − 0.35) but a positive relationship among full siblings (r = 0.28). The relationship between parent and offspring lifetime reproductive success (LRS) and full sibling LRS was strongly and positively correlated across time (r = 0.799, p < 0.001). Similarly, the LRS among full siblings was positively correlated with both the LRS among half siblings (r = 0.532, p = 0.011) and the relationship between the LRS of aunts and uncles with their nieces and nephews (r = 0.438, p = 0.042).

Discussion

We show that an individual’s lifetime reproductive success is best predicted by the reproduction of their full and half siblings, but not their parents, grandparents or aunts and uncles. Because all siblings share at least one parent, we believe parental investment has had an important impact on fitness. Overall, these results indicate that direct parental investment, but not genes, is likely to have had an important and persistent impact on lifetime reproductive success across more than two centuries of Icelandic history.

Introduction

Parental investment (PI) is crucial for the success and survival of offspring in many species (Trivers, 1972). Direct parental investment may vary in form, amount, and duration, which can affect offspring survival, growth, and development. To tease apart direct parental investment from genetic contributions, heritability estimates can be used to measure the respective effects that genes and the environment have on a given phenotype. These proportions consider the fraction of the phenotypic variation between individuals in a population that results from genotypes (Falconer, 1989; Raj & Van Oudenaarden, 2008).

There are several ways to estimate heritability. The simplest method is to either compare parent and offspring traits (the mid-parent–offspring regression) or compare the similarity of traits between full siblings (full sibling’s correlation). Higher correlations between the traits of family members suggests higher heritability (Visscher, Hill & Wray, 2008; Visscher et al., 2006). These estimates, however, represent an upper limit for the effect of genes on a trait and assume there is no environmental effect (Falconer, 1989). It should be noted that heritability estimates are also biased by sources of non-additive genetic variance (e.g., dominance and epistasis). For example, heritability estimates obtained from parent–offspring and full-sibling regressions include dominance variance, while this is not the case for estimates obtained from half-siblings.

In the past, the most common technique for estimating heritability in humans has been the twin design (Lynch & Walsh, 1998) while the “animal model”, in which a pedigree is entered into a restricted maximum likelihood (REML) model, has gained popularity over recent decades for non-human species (Henderson, 1975; Hill, Goddard & Visscher, 2008; Visscher, Hill & Wray, 2008; Wilson et al., 2010). Animal models tend to yield slightly lower heritability estimates than those obtained from parent–offspring regressions because they are less biased by common environments shared by first order relatives (i.e., parent–offspring and full sibling dyads) (Åkesson et al., 2008). Different methods introduce different biases and the impact of common environments inflating heritability estimates are of particular concern when studying humans because we often share complex and overlapping environments with close relatives (Visscher, Hill & Wray, 2008). More recently, detailed genetic data can be used to sidestep these issues and produce estimates unbiased by shared environments. By taking advantage of the random segregation of parental alleles (e.g., full siblings share a mean coefficient of relatedness of 0.5 but exhibit considerable variance around this mean (SD ± 4.0%)), researchers can generate extremely accurate estimates of both environmental and genetic effects (Visscher, Hill & Wray, 2008).

Table 1 Correlation results for lifetime reproductive success across kin types.

Relationship between lifetime reproductive success of different categories of first and second degree relatives for all individuals.

	Full siblings (all)	Full siblings partial (controlling for birth interval)	Parents (all)	Half siblings	Aunts and uncles	Grandparents	
Lifetime reproductive success	0.191	0.21	0.00	0.07	0.03	0.01	
	(p < 0.001)**	(p < 0.001)**	(p = 0.08)	(p < 0.001)**	(p < 0.001)**	(p < 0.01)*	
	N = 153,084	N = 153,084	N = 163,081	N = 74,129	N = 147,912	N = 158,571	
Notes.

* p < 0.01.

** p < 0.001.

Figure 1 Lifetime reproductive success among first degree relatives.

The heritability estimates for LRS for first degree relatives (r = 1∕2); (among full siblings and between parents and offspring) were positively correlated across decades.

Although heritability estimates can be good for measuring the relative contribution of genes and environments, they are often confounded when environments are shared by kin. This is because differences in life history outcomes between individuals, especially close relatives, can result from both environmental and genetic effects (Fisher, 1930; Kosova, Abney & Ober, 2010). Furthermore, heritability estimates obtained by comparing the traits of relatives are differentially and upwardly biased, depending on the degree to which the relatives share common environments.

Heritability estimates obtained from the genotypes of relatives, such as full siblings, can therefore be used to identify the precise proportion of a trait resulting from additive genetic variance. In other words, we can use genetic data to determine exactly how much of a given phenotype results from shared genes and how much is due to shared environments. Meanwhile, comparing traits between different classes of relatives (e.g., among siblings or between parents and offspring) can help to partition the remaining environmental variance into various components. Therefore, the degree to which estimates between different categories of relatives are biased can be used to provide insight into which environmental factors (e.g., parents, common household and geographic location) have the largest effects on a given phenotype. In sum, comparing traits between relatives who share a different degree of genetic relatedness and different types of environments can help partition these environmental effects.

The aim of this study was to determine the impact that parents have on the lifetime reproductive success (LRS) of their children. To date, a small number of studies have been able to use large genealogical databases to estimate the heritability of reproductive success. One study of a late 19th century population from Utah showed that increasing parental reproduction reduced the reproductive success of offspring (i.e., the more siblings that individuals had the lower their LRS) (Penn & Smith, 2007). High heritability’s were obtained for several key life-history traits (e.g., fecundity, interbirth interval, age at last reproduction, and adult longevity) from an animal model for females, but not males, in a preindustrial Finnish population in one study (Pettay et al., 2005) and for both sexes in another (Bolund et al., 2013). These estimates were also seen to persist after the demographic transition (after 1880 in Finland) (Bolund et al., 2015). Another study of alpine communities in Tyrol, Italy found moderate to low heritability of LRS (h2 ∼ 0.07) for both males and females, but these estimates did not consider the shared environmental effects of a common household or shared parents (Gögele et al., 2011). Finally, age at first reproduction of females in a pre-industrial French Canadian population was shown to decline by 4 years over a 140-year period and heritability estimates derived from an animal model indicated that this change was likely to have arisen from genetic changes (Milot et al., 2011).

Using an exceptionally accurate and large database, spanning over two centuries, we examined the relationship between parental LRS and offspring LRS. Because parents and offspring share half their genes, we predicted that if genes have had an important effect on reproduction, then the LRS of ones’ parents will predict an individual’s LRS. Meanwhile, if parental investment (PI; cf Trivers, 1972) and the environment provided by parents (e.g., a common household) have had a stronger impact on reproductive success, then full siblings will predict ones LRS. This is because, in addition to sharing the same average degree of relatedness as parents and offspring (r = 0.5), full siblings also share parents. Furthermore, we predicted that if these relationships are strongly influenced by the environment that parents provide (e.g., a common household) they will respond to environmental volatility and should change over time. This study examined the effects that different categories of first (full siblings and parents) and second degree relatives (half siblings, grandparents and aunts and uncles) have had on an individual’s LRS across over two hundred years of Icelandic history.

Materials and Methods

The database

We used an Icleandic genealogy, also known as Íslendingabók, to test our predictions. Íslendingabók has several important advantages over other genealogical databases. First, it is a population-based database that contains information about approximately 600,000 of the one million individuals estimated to have ever inhabited Iceland (Gudhmundsson et al., 2000). A population-based genealogy is advantageous because there is less of a chance of imposing a sampling bias. Using a national census to generate demographic information, reduces the possibility of over or under report any particular group (Moorad et al., 2011). Second, the database is extremely accurate as it includes all living Icelanders and most of their ancestors. An examination of mitochondrial DNA shows a maternal accuracy rate of 99.3% (Siguradottir et al., 2000) while the mistakenly assigned paternity error rate is estimated to be less than 1.5% (Gudhmundsson et al., 2000). A third advantage of Íslendingabók is that it is based on a relatively isolated population. This allows for a more accurate genealogical analysis across multiple generations. Finally, because Iceland’s economy did not undergo industrialization until after World War II, there are extensive life history data on individuals who reproduced prior to the massive changes in mortality and fertility rates that characterize the demographic transition—the process by which high fertility and mortality rates are replaced by low fertility and mortality rates (McNicoll, 1992).

The first national census in Iceland was taken in 1703 (Karlsson, 2000), and these data were often regarded as the most accurate (K Gundarson, pers. comm., 2011). Therefore, all our analyses were restricted to individuals born after 1700. Our analyses were also limited to individuals who were born before the industrialization of Iceland, circa 1920. This was done for two reasons. First, the demographic transition may have fundamentally changed patterns of population growth that had been more typical of ancestral human populations (Barthold, Myrskylä & Jones, 2012). Second, the expected positive relationship between resources and reproduction is often decoupled after the demographic transition, whereby poverty is associated with high fertility rates (McNicoll, 1992).

Fertility data

Individual LRS was the primary dependent variable, but mean fertility rates fluctuated considerably across time. For example, average fertility rates for individuals born between 1810 and 1819 peak at 3.12 children but drop to 1.98 only thirty years later (1850–1859). Therefore, all LRS values for all individuals were first log transformed and standardized by the decade in which the individual was born ([log transformed value − mean of decade]/ standard deviation of decade). This transformation made the reproduction of individuals less dependent on changes in mean family size across birth decades, so that effective comparisons could be made across time (Helgason et al., 2008).

Heritability estimates among relatives

Although heritability estimates based on precise coefficients of relatedness among full siblings can be used to obtain accurate and unbiased estimates of the genetic impacts of parents, distinguishing the effect of parental investment (PI) from other environmental factors (e.g., shared household, community or geographic region) is more difficult. One way to isolate the effects of PI on offspring LRS, however, is to compare the reproductive similarity of close relatives across varying degrees of social and genetic relatedness. Parents, children, and full siblings, for instance, all share a coefficient of relatedness (r) of 0.5 and are all first-degree relatives while half-siblings, aunts-uncle and nieces and nephews, and grandparents and their grandchildren are all second-degree relatives and share an average coefficient of relatedness (r) of 0.25. Comparing the degree to which different types of relatives with the same coefficients of relatedness share fitness outcomes may yield insights into the impacts that different environments have on reproduction.

For example, although members of the immediate family are all first-degree relatives and are therefore all related by the same coefficient of relatedness, the environments they share are different. For example, full siblings share parents, while parents and their children do not. Meanwhile comparing relatives with different coefficients of relatedness who often share similar environments (e.g., full siblings (r = 1∕2) and half siblings (r = 1∕4)) may also produce insights. Therefore, comparing different types of relatives both between and within different categories of genetic and social relatedness may shed light on the effects of sharing different types of environments on an individual’s LRS. This is because heritability estimates based on correlations between relatives depend critically on the assumption that the environments shared between relatives with the same mean coefficients of relatedness are the same across all social categories and it is often unclear which relatives share a common environment. For example, do parent–offspring pairs have the same amount and type of environmental sharing as siblings? (Zaitlen et al., 2013). In short, if different types of relatives exist in different environments, heritability estimates will be biased (Thomas, Pemberton & Hill, 2000; Wray & Visscher, 2008; Zaitlen et al., 2013).

In this study, broad sense heritability estimates (H2) for LRS were obtained from the historic genealogical data by correlating the traits of two types of first degree relatives (coefficient of relatedness, r = 1∕2): parents with their children and full siblings. Parent-offspring heritability was measured by correlating the mid-parent standardized reproduction or lifespan (e.g., [mothers LRS + fathers LRS]/2) with the standardized reproduction of their offspring. Full sibling heritability was measured by correlating an individual’s standardized LRS with the mean of the standardized LRS of all his or her full siblings. Correlations of the LRS of individuals with second degree relatives (r = 1∕4; aunts and uncles, grandparents, and half siblings) were also obtained. All correlations between the LRS of relatives were calculated as the proband’s standardized LRS to the mean of the standardized LRS of all his or her relatives in the comparison group (e.g., mean LRS of all aunts and uncles). All correlations were weighted by family size to provide equal weight to each group of relatives. For example, in determining the correlation amongst full siblings, individuals who had 4 full siblings were weighted 1/4 as much as those who had only one. Using SPSS, the ‘weight cases by’ command was used, the frequency of individuals for each type of relative was entered.

Simulations

Three models were constructed in Visual Basic to simulate the impact of parental investment and genes on offspring reproduction. Specifically, the models were designed to simulate the effects of parents transferring a finite amount of investment (i.e., that which is diluted by additional offspring) called ‘resources,’ and or a fixed (i.e., undiluted by one’s number of offspring) effect which we called ‘genes.’ These results were then compared with the actual results from the genealogical data to gain insight into which model best approximated the  data.

Model 1, The Null Model: This model assumed neither genes nor resources played any role in the number of offspring an individual produced. Reproduction followed a random Poisson distribution and was based solely on certain fixed parameters, including the mean number of offspring per generation and the standard deviation of each generation’s size.

Model 2, The Resources Model: Here, the number of children, and hence grandchildren, that an individual produced was tied to the number of resources that the F1 generation transmitted to their offspring.

Model 3, The Genetic Model: This model was based on Mendelian inheritance, in which the number of children and grandchildren produced was determined by the alleles they inherited from each of their parents.

Model 4, The Genes and Resources Model: A simulation was used to explore the impact of both genes and resources on an individual’s reproduction.

All reproductive values (LRS) were log transformed and standardized [(log transformed value—mean of decade)/ standard deviation of decade)] by the individual’s birth decade. Pearson correlations on the y-axis are between mid-parent (mother and fathers LRS average) and offspring LRS and among full siblings (each siblings LRS was correlated with the average of all their full siblings) for each decade shown on the x-axis (1700–1919). Correlations were weighted by number (e.g., if an individual had either 10 full siblings or 10 children they each were each statistically weighted at 0.1).

Results of the resource model and the genetic model were based on 200 simulations (Table 2). There were 1,000 parents (F1 generation) in each simulation with an average reproduction per generation of two and a genetic fitness set equal to one. The resources available in the F1 generation had a mean of one with a standard deviation of two and the mean number of children per generation was set equal to two.

Family size varied widely across and within decades and because individuals within a particular family were not independent, larger families were expected to inflate correlations. We addressed this problem by randomly choosing the reproduction of just two children (e.g., the first was equally as likely to be chosen as the 4th). This method ensured larger families were not contributing more measurements in the regressions and within group correlations than smaller families. For the mid-parent (the average of the mother and fathers LRS) offspring correlations, we regressed the average number of children (F2 generation) from each set of parents on the average number of grandchildren of two randomly selected children (F3 generation). This process of randomly selecting two individuals and averaging their reproduction eliminated the erroneous inflation of both the full sibling and parent–offspring correlations from the results of the simulations. This same problem was solved for the genealogical data by standardizing LRS within everyone’s birth decade and statistically weighting the data by the number of individuals in each class of relatives (e.g., half siblings, number of grandchildren).

Genetic data

Genetic data for some individuals who were alive after 1996 were available from deCode genetics and were based on approximately 300,000 single nucleotide polymorphisms (SNP’s) for each genotyped individual. Therefore, accurate coefficients of relatedness were obtained for 8,456 pairs of full siblings and were entered into a restricted maximum likelihood (REML) model. The lifetime reproductive success of these individuals was standardized by sex, birth year and geographic region. This was only done for the genotyped individuals, however, because geographic region was not available for individuals in the historic dataset. Full siblings are related, on average, by a coefficient of relatedness of 1/2. Due to the process of meiosis, however, there are small but significant differences in coefficient of relatedness between them. For example, one pair of full siblings may be related at r = 0.42 while another pair is related at r = 0.58. These differences in relatedness between full siblings can be used to obtain unbiased narrow sense heritability estimates for any trait. We used a dummy variable (1 = full siblings and 0 = all other pairs), to estimate a ‘family effect.’ In other words, we could distinguish the effect of full siblings, due to their being members of the same family and living together (i.e., an environmental effect), from the effects of genes that full siblings share identical by descent. Because heritability estimates can fluctuate considerably over time as the overall environmental impact on a given phenotype changes, these estimates only provided us with an accurate heritability estimate of LRS for contemporary Iceland.

Results

The database (Íslendingabók)

Analysis of the genealogical data from Íslendingabók for all individuals born between 1700 and 1920 revealed that LRS was a strong predictor of the number of grandchildren an individual produced (r = 0.63, p < 10−6). LRS was therefore considered to be a reasonable proxy for evolutionary ‘fitness.’

The overall relationship between the reproduction of all full sibling pairs, across all birth years (1700–1919) was significant and positive, r = 0.19 (p < 0.001) while there was no detectable relationship between the reproduction of parents and their offspring, r = 0.00, p = 0.08 (Table 1). Nevertheless, across decades the relationship between parent and offspring LRS is correlated with the LRS correlation among full siblings (r = 0.799, p < 0.001; Fig. 1). In other words, the regressions are positively associated such that when the LRS of parents and offspring is more correlated the relationship among full siblings is also more correlated. Similarly, the correlation of LRS among full siblings was positively correlated with both the LRS among half siblings (r = 0.532, p = 0.011; Fig. 2) and the relationship between the LRS of aunts and uncles with their nieces and nephews (r = 0.438, p = 0.042; Fig. 3) across decades. On the other hand, there was no detectable relationship between the correlations with one’s grandparents LRS and any other category of relative examined (see Table S1). Nor was there any significant association between one’s reproductive similarity with aunts and uncles and the reproductive correlation with one’s parents across decades (r = 0.166, p = 0.46).

Figure 2 Lifetime reproductive success among full siblings and half siblings.

The correlations of LRS among full siblings (r = 1∕2) and those among half siblings (r = 1∕4) were positively correlated across decades.

Figure 3 Lifetime reproductive success among full siblings and between aunts, uncles, nieces, and nephews.

The correlations of LRS among full siblings (r = 1∕2) and between aunts and uncles and their nieces and nephews (r = 1∕4) were positively correlated across decades.

Table 2 The relationship between birth interval on the reproduction of full siblings.

	Resources (s.e)	Genes (s.e)	Genes and resources (s.e)	
Parent-offspring	r =  − 0.35 (0.02)	r = 0.185 (0.03)	r = 0.06 (0.03)	
Full siblings	r = 0.28 (0.03)	r = 0.20 (0.03)	r = 0.21 (0.02)	

There were major demographic changes that also interacted with these heritability estimates. Average reproduction per decade was negatively associated with the reproductive correlation among full siblings (r =  − 0.603, p < 0.001) and the correlation between parents and offspring (r =  − 0.44, p = 0.04). So as population wide reproductive rates increased, both full sibling and parent–offspring correlations declined. Population was also positively, but not significantly, correlated with the parent–offspring correlation (r = 0.347, p = 0.11). Finally, birth interval was weakly, but significantly, correlated with LRS (Pearson’s R = 0.041 (p < 0.001). In other words, full siblings who were closer in age had more similar fertility.

Simulations

Results of the resources model, in which resources that are transmitted from parents to children and distributed equally amongst siblings affects reproductive success, revealed a negative relationship between parent and offspring reproduction but a positive one among full siblings (Table 2). The genetic model, in which offspring inherit their parent’s reproduction with some probability, showed nearly identical correlations between the LRS of parents and offspring and those among full siblings. The full siblings and mid-parent–offspring correlations in LRS are both within two standard errors (95% confidence intervals) of the models simulating “genes and resources,” and in a maximum likelihood model the genes and resources model also provides the best fit for the genealogical data (e.g., produces the lowest Akaike Information Criterion score).

Genetic data

A restricted maximum likelihood model (REML) entering the precise coefficients of relatedness for 8,456 pairs of full siblings yielded a narrow sense heritability estimate (h2) of 0.137 with a standard error of (0.02). Adding a family effect (i.e., identifying the pairs of full siblings) to the model and allowing it to compete with these coefficients of relatedness, however, revealed a family effect (f2) of 0.129 (0.03) and a genetic effect of 0.00 (0.05) This suggests that the heritability estimate (h2 = 0.137) was based solely on shared family effects among full siblings and was not due to shared genes. Furthermore, the intra-class correlation of LRS among the 8,456 pairs of full siblings was r = 0.076 (h2 = 0.152) reinforcing results from the REML model and indicating that the heritability estimate of LRS in modern Iceland may be lower than in previous centuries.

Discussion

Analysis of a genealogical database across 220 years of Icelandic history (1700–1920) showed that an individual’s lifetime reproductive success (LRS) is strongly predicted by the reproduction of their full siblings but not their parents (Fig. 1, Table 1). This study also revealed a negligible heritability (h2) for LRS, but a significant shared family effect (f2) of parents on the reproduction of their children for the current population based on results from a restricted maximum likelihood model using precise IBD gene sharing (SNPs) amongst full sibling pairs. Finally, results of a model simulating the effects of genes and resources on LRS were most like the overall results from the genealogy (see comparison of Tables 1 and 2). Overall, these results suggest that parental investment (PI) has had an important effect on the LRS of children across much of Iceland’s history.

Why do individuals reproduce like their siblings, but not their parents? Siblings bonds in humans are particularly strong, and brothers and sisters often maintain relationships throughout life (Rodseth et al., 1991; Chapais, 2011). In hunter gather societies, for example, brothers and sisters commonly co-reside into adulthood (Hill et al., 2011). During the 18th and 19th century in Iceland, children often shared the same household with their parents after they married (Wall, Robin & Laslett, 1983; Moring, 2003). Therefore, the environments that are shared by siblings throughout development and often past sexual maturity may contribute to similar fertility rates.

An individual’s LRS is best predicted by the reproduction of full siblings, followed by half siblings, aunts and uncles, grandparents and finally is least like their parents. The relationship amongst the LRS of half siblings who only share one parent is slightly less than half of the relationship among full siblings who share both parents. This is consistent with the view that the environment parents provide, either through direct investment or the effects of a shared household, has an important impact on one’s reproduction because full and half siblings share parents, but parents and offspring do not. Meanwhile, the overall undetectable relationship between parent and offspring LRS suggests a low heritability of this trait. The unbiased estimates from the SNP data support this result and suggest a very low h2 in the current population. Although most previous estimates of heritability’s using genealogies have shown a more moderate heritability (∼0.10) of LRS, these estimates have often been obtained using an animal model in which an entire pedigree is used and are therefore not able to parse specific intra-familial relationships. The low heritability estimates obtained in our study are consistent with Fishers fundamental theory of natural selection which states that additive genetic variance (i.e., narrow sense heritability) for traits closely tied to fitness should be close to zero in natural populations at equilibrium (Fisher, 1930; Falconer, 1989). Empirical data from many animal populations also supports this conclusion (Gustafsson, 1986). The observation that this relationship is negative in many decades, however, indicates that there may be a quantity-quality tradeoff for parents. Consistent with this interpretation, each successive child that a parent produced, both shortened the lifespan and lowered the reproduction of all other offspring (Lynch, 2016). Furthermore, an individual’s reproduction is not expected to be either negatively or strongly positively affected by the LRS of their full aunts and uncles. This is because aunts and uncles are not expected to invest as heavily in their nieces and nephews as they would their own children. In 18th and 19th century Finland, for example, the presence of non-reproductive aunts and uncles was weakly or negatively correlated with the survival of their nieces and nephews (Nitsch, Faurie & Lummaa, 2014). Therefore, unlike the parent–offspring relationship, niece and nephews are less likely to be negatively impacted by the dilution of investment (e.g., a negative correlation in LRS is evidence of a quantity–quality tradeoff between offspring number and offspring fitness). Therefore, these within family patterns are all broadly consistent with the interpretation that parental investment has had an important impact on fitness in Iceland across the 18th and 19th centuries.

The observation that the heritability estimates from the correlations between parents and offspring and those among full siblings are positively correlated across time suggests that these two estimates may depend on similar factors. For parents, the amount of investment available per child is diluted when additional offspring are produced. This can generate a negative relationship between parent and offspring reproduction. In contrast, investment is often shared among full siblings which can generate a positive association among siblings. The strong correlation between these two estimates, however, indicates that when PI increases, the quantity-quality tradeoff decreases. If parents increase overall investment in offspring, it makes sense that both correlations should increase. Siblings will share a higher level of investment and will therefore reproduce even more similarly, while the quantity-quality tradeoff will also be reduced because overall investment is higher and additional siblings will cost less. The significant longitudinal relationships between the LRS of other relatives (full siblings with half siblings (Fig. 2) and full siblings with aunt and uncles (Fig. 3) also supports this interpretation.

These relationships also appear to respond to mortality shocks. Two peaks of the parent–offspring curve coincide with the two most significant population declines in Iceland between 1700 and 1919: the smallpox epidemic of 1707–1709 with an estimated mortality rate of 26% (measurement based on all births 1707–1709) (Adalsteinsson, 1985) and the Laki volcanic eruption of 1783 during which approximately 1/4 of the population and 1/2 of the livestock died (Jackson, 1982; Thorarinsson, 1961). Survivors of these events may have had more opportunities to acquire resources as the population declined and farmland became available. In Iceland marriage (and hence reproduction) was often tied to owning a farm (Karlsson, 2000). Therefore, as population density declined new opportunities may have been created which allowed the birth cohort in the following years to be more successful. In addition, parental investment may be less important when population, and therefore competition, is lower. The negative relationships between population wide reproductive rates and both heritability estimates derived from first degree relatives (i.e., parents to offspring and full siblings) provides support for this view. There is also a positive, albeit non-significant, relationship between the population of Iceland and the parent–offspring regression. In other words, when the population increases, the quantity-quality tradeoff declines (i.e., siblings are less costly to one’s fitness). These results are consistent with a study of a lizard population which revealed a cyclical relationship between population density, reproductive effort and investment such that morphs producing few, larger eggs (K selection) are favored at high population densities while those that produce more, but smaller eggs (R selection) are favored at low densities (Sinervo, Svensson & Comendant, 2000).

There are reasons to be careful when attempting to interpret demographic changes across multiple generations, however. These fluctuations are a complex and dynamic interaction of many factors. For instance, the assumption that wealthier parents should achieve higher LRS (a central tenant of sociobiology) (Nettle & Pollet, 2008), is often violated in contemporary societies where an inverse relationship is now usually seen between family size and wealth (see Vining, 1986). Unfortunately, there were no direct measures on income or material resources for any individuals in the genealogy. Prior to the demographic transition, however, wealthier families probably had more children (Kirk, 1996). If true, this would only serve to reduce the effects observed here because the dilution of resources is reduced when wealthy families have more children. Therefore, these results are likely to be conservative. This may also be one reason why the genealogy often resembles the ‘genes and resources’ model more than the ‘resources only’ model. Another reason may be that as Iceland rises above a subsistence level economy in the late 19th and early 20th centuries and resources become more plentiful, the variance of environmental factors which contribute to the denominator in h2 decreases. This would also help to explain previous research revealing higher correlations between the LRS of parents and their offspring in resource rich environments (Tuvblad, Grann & Lichtenstein, 2006; Garcia de Leaniz & Consuegra, 2006; Charmantier & Garant, 2005; Sgro & Hoffmann, 2004; Silventoinen, 2003; Merila, 1997).

This study demonstrates that an individual’s LRS is best predicted by the reproduction of their full and half siblings. These correlations fluctuate over time which suggests that fertility rates and PI are sensitive to changes in the environment. Future studies can build on these methods and results, expanding both our understanding of the heritability of fitness and the impact of parents. For example, carefully examining the different environmental influences on LRS may help to distinguish between the effects of different types of relatives and common households from those of peer groups. In addition, expanding this analysis to other populations may enhance our understanding of the importance of sibling bonds, while also further teasing apart the root causes of sibling influence on fitness components. Overall, however, these results indicate that parental investment and or resources have had an important impact on the reproduction of offspring in Iceland.

Supplemental Information

Supplemental Information 1 Summary data for all first and second degree relatives used in Figs. 1, 2 and 3 and Table 1

Click here for additional data file.

We are grateful to Kári Stefánsson for access to the database and we thank Agnar Helgason for his assistance with the data analysis.

Additional Information and Declarations

Competing Interests

Author Contributions

Data Availability

The authors declare there are no competing interests.

Robert F. Lynch conceived and designed the experiments, performed the experiments, analyzed the data, contributed reagents/materials/analysis tools, wrote the paper, prepared figures and/or tables, reviewed drafts of the paper, grant recipient.

Emily C. Lynch reviewed drafts of the paper.

The following information was supplied regarding data availability:

The summary data on which these analyses are based are available in Table S1. The raw data upon which these analyses are based, however, are proprietary and were obtained from a third party at the offices of deCode genetics in Reykjavik, Iceland. Access to the data can be requested from Agnar Helgason at email: Agnar.helgason@decode.is. The data is proprietary and contains sensitive genetic information on the people of Iceland. Due to both privacy and ownership (of the database) issues, deCode will not allow researchers to take an electronic copy of the database off site. Therefore, all analysis must take place at the offices of deCode genetics in Reykjavik, Iceland.

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
