# Peer review of "The impact of parental investment on lifetime reproductive success in Iceland"

_PeerJ, doi:10.7717/peerj.2904_

## Round 0.1 · original submission · Major Revisions

Please address all the suggestions the reviewers have made to improve your manuscript, and try to clarify the points that the reviewers have found difficult to follow

Reviewer 1 ·

Basic reporting

This paper presents interesting analyses of fertility data from Icelanders born 1700 to 1920.

I had a bit of trouble understanding the study and results at first, because the write-up was not very clear. This is on both the level of sentences and the broader level of emphasising the important things and de-emphasising details. For example, in the abstract the results reported in the first two sentences are both very difficult to understand (based as they are on associations between associations, or associations between an association and time), and also are not the most relevant of the paper. In fact, by far the most striking and important result of the paper, which is the zero parent-offspring correlation for fertility, is not even mentioned. A similar pattern exists in the results, where seemingly irrelevant statistics are jumbled in with important ones, and the main story is obscure.

Anyway, once I managed to understand what the data were saying and its relevance, I found it really interesting. Particularly surprising, as I said, is the zero parent-offspring correlation in fertility. This contrasts with other studies, which should be dealt with at least in the discussion, and perhaps in the intro as well. The study is poorly contextualised in the extant literature on the topic. For example, what other heritability estimates have been found, using what methodologies, in what countries and eras? What about any other looks at parent-offspring correlations, or quantity-quality tradeoffs? What should be made of the differences? etc.

I think the paper has potential, but needs work.

Experimental design

No comments

Validity of the findings

Some strange choices were made in analysis, and more transparency/detail around these decisions would help the reader to judge for themselves the validity of the findings.

Reviewer 2 ·

Basic reporting

No comments, except for one missing reference in the reference list (details in general comments to the author)

Experimental design

No comments

Validity of the findings

Some major concerns regarding the statistical analyses of the data, detailed in the general comments to the author.

Additional comments

Major comments
I have reviewed this manuscript previously for another journal. While I do think that this manuscript has a lot of potential and I am now happy with the theoretical reasoning and framing of the study, I have some major remaining concerns regarding the analyses, and these are detailed below.

Abstract
The abstract could be more clear, for example, analyses of heritability are introduced in the methods part of the abstract, but no results mentioned. Conversely, the resources model is mentioned in the results part of the abstract without any kind of introduction/explanation of what it is and what it is testing.

L61-66. Repetition of paragraph above

L67-73. This is the key point

L77-78, l86-88, the goal of the study is here defined in 2 different ways.

L141-146. This crucial statement is not supported by the cited references. Griffiths et al seem to be missing for the reference list, so I did attempt to find it, but while both Wray & Visscher and Zaitlen et al. point out that common MZ/DZ twin methods do assume that ’the two classes of siblings share the same relevant environmental exposures’ (to paraphrase Zaitlen et al, discussion), this is however NOT the case for other methods (that make use of half-siblings, aunts and uncles). For example: The animal model divides the total phenotypic variances into components, of which the genetic component is often the main focus. However, different parts of the environmental variance can also be decomposed in the model, for example, maternal effects are often studied in animal models. Other aspects of environmental variance can also be disentangled. It is thus a crucial assumption of the model that individuals of the same relatedness category do NOT share the same environment. If they did, the model would not be able to disentangle genetic from environmental components of the variance because they would be completely confounded. This is in fact a common criticism of the animal model framework when applied to humans, because in human pedigrees, relatives tend to also share a common environment to perhaps a higher degree than in many other species. The model thus makes use of cases like half-siblings and aunts and uncles, that is individuals that share the same relationship level but that share a common environment to different degrees, to be able to disentangle genetic from environmental influences on traits.

L147-161. I am sorry about this harsh comment, but why are you doing this? You estimate unbiased narrow-sense h2 estimates from genetic data, so why are you doing these analyses? You look at different types of relatives to disentangle the environmental contribution that they have on their offspring LRS, but why are also you using them to estimate h2, if you are getting unbiased estimates from the genetic data? The h2 estimates from the different sets of relatives will all be biased to different degrees, by shared environments, dominance, epistasis (see eg Zaitlen et al, and as you have discussed at length in the introduction). Also, these correlations between relatives would yield broad-sense, not narrow-sense, h2 estimates (l147), precisely because they include other parts of the genetic variance in addition to the additive genetic variance. Please clarify your reasoning, if you use these different correlations mainly to disentangle different aspects of the environmental variance, this could, as suggested above, be done in one step in an animal model frame work.

L151-161. What types of correlations are these? Is it Pearson correlations? Why not instead use generalized linear mixed–effects models and estimate heritability from the slope of the line (beta) as is commonly done. This would allow to control for covariates in the same model (eg decade).

L163. The simulations are not introduced in any way to justify what their purpose is.

L212. Heritability estimates of what, I presume LRS?

L145. I cannot find Griffiths et al in the reference list

Footnotes: it would be easy to remove the footnotes and either include the information in the running text, or refer to it by citing the reference.

---

## Round 0.2 · Minor Revisions

Please address the issues raised by the reviewers

Reviewer 1 ·

Basic reporting

While the clarity has improved in this revision, some things are still poorly described. For example, the sentence starting 'Nevertheless,..' on ln256 reports a main finding but does not make sense (the relationship between X and Y and Z was correlated across time - doesn't specify what is correlated with what). The next sentence has a similar problem.

It is claimed that the variance of environmental factors is the denominator of h2 (ln376), but this is wrong. The denominator they are referring to should be total variance, not environmental variance.

There are also grammatical errors throughout, especially missing possessive apostrophes and inappropriate commas.

Experimental design

None

Validity of the findings

One concern I had with the results was with the REML analysis. Given that the REML analysis of full sibling pairs is unbiased and unconfounded with shared environmental effects, how can the h2 estimate go from 13.7% to zero when family effects are modelled too? This doesn’t make sense to me. To me, the 13.7% number should be accurate, whereas the authors interpret the zero number instead.

Reviewer 2 ·

Basic reporting

No comments

Experimental design

No comments

Validity of the findings

No comments

Additional comments

This is the second time a review this manuscript for this journal (reviewer 2 in the previous round). The authors have done a thorough job of responding to the reviewers’ comments. They have done requested changes and clarified points of confusion. The analytical process is now also more clear and justified. I still have a few comments regarding the introduction and discussion, detailed below.

Major comments

L56. You might want to also discuss how heritability estimates are not only biased by environmental variance but also by sources of non-additive genetic variance (dominance, epistasis) and this varies between methods (for example: estiamtes of heritability obtained from parent-offspring and full-sib regressions includes dominance variance, while this is not the case for estimates obtained from half-siblings).

L89-90. There are several more studies that have used genealogical data to estimate heritability of reproductive success, or other fitness proxies, like intrinsic rate of increase, in humans.

Some examples on women:

Milot E et al. 2011. Evidence for evolution in response to natural selection in a contemporary human population. Proc. Natl Acad. Sci. USA

Pettay et al. 2005. Heritability and genetic constraints of life-history trait evolution in preindustrial humans. Proc. Natl Acad. Sci. USA

On both sexes:

Bolund et al. 2013. Divergent selection on, but no genetic conflict over, female and male timing and rate of reproduction in a human population. Proc. R. Soc. B

Bolund et al. 2015. Effects of the demographic transition on the genetic variances and covariances of human life-history traits. Evolution

Gögele et al. 2010. Heritability Analysis of Life Span in a Semi-isolated Population Followed Across Four Centuries Reveals the Presence of Pleiotropy Between Life Span and Reproduction. Journal of Gerontology.

L312-313. While the correlation between parents and offspring was estimated at zero, the positive correlations between all other pairs of relatives suggest a non-zero heritability, and there is no reason to draw conclusion regarding h2 based on the parent-offspring correlation alone, so it is difficult to draw any firm conclusions about the h2 of LRS in the historic data, given that the estimates from the different sets of relatives a) vary and b) are biased to different degrees. However, the unbiased estimation from the SNP data also suggest a very low h2 in the more modern sample, so this would be worth discussing here. These findings could also be discussed d in relation to previous estimates of h2 of LRS in other human populations, see earlier comment about previous findings.

L322. “aunts and uncles do not normally distribute resources”. This is not backed up by any reference, it would be nice if there is specific information for this population regarding resource transfer/investment from these more distant classes of relatives to support this claim. For example, one can easily envision how childless aunts and uncles in preindustrial societies provided support and eventually an inheritance to their childbearing relatives, but this remains speculation if there are no supporting references.


Minor comments

L 46-50. I suggest re-writing this slightly: ”heritability estimates can be used to measure the respective effects that genetic variance and the environment have on a given phenotype. These proportions consider the fraction of the phenotypic variation between individuals in a population that results from genotypes”

L67. Remove one ’of’

L69. Remove ’the variance’

L92. Remove ’s’

---

## Round 0.3 · accepted · Accept

Reviewers' comments have been satisfactorily addressed